# Reduced Rank Regression-Derived Dietary Patterns Related to the Fatty Liver Index and Associations with Type 2 Diabetes Mellitus among Ghanaian Populations under Transition: The RODAM Study

**DOI:** 10.3390/nu13113679

**Published:** 2021-10-20

**Authors:** Tracy Bonsu Osei, Anne-Marieke van Dijk, Sjoerd Dingerink, Felix Patience Chilunga, Erik Beune, Karlijn Anna Catharina Meeks, Silver Bahendeka, Matthias Bernd Schulze, Charles Agyemang, Mary Nicolaou, Adriaan Georgius Holleboom, Ina Danquah

**Affiliations:** 1Heidelberg Institute of Global Health (HIGH), Faculty of Medicine and University Hospital, Heidelberg University, 69120 Heidelberg, Germany; tracy.osei@uni-heidelberg.de; 2Department of Vascular Medicine, Amsterdam UMC, University of Amsterdam, 1105 AZ Amsterdam, The Netherlands; a.vandijk2@amsterdamumc.nl (A.-M.v.D.); s.dingerink@amsterdamumc.nl (S.D.); a.g.holleboom@amsterdamumc.nl (A.G.H.); 3Department of Public Health, Amsterdam UMC, University of Amsterdam, 1105 AZ Amsterdam, The Netherlands; f.p.chilunga@amsterdamumc.nl (F.P.C.); e.j.beune@amsterdamumc.nl (E.B.); k.a.meeks@amsterdamumc.nl (K.A.C.M.); c.o.agyemang@amsterdamumc.nl (C.A.); m.nicolaou@amsterdamumc.nl (M.N.); 4Center for Research on Genomics and Global Health, National Human Genome Research Institute, Bethesda, MD 20892-2152, USA; 5Department of Internal Medicine, St. Francis Hospital Nsambya, MKPGMS-Uganda Martyrs University, Kampala 5498, Uganda; silverbahendeka@gmail.com; 6Department of Molecular Epidemiology, German Institute of Human Nutrition Potsdam-Rehbruecke, 14558 Nuthetal, Germany; mschulze@dife.de; 7Institute of Nutritional Science, University of Potsdam, 14558 Nuthetal, Germany

**Keywords:** dietary pattern, reduced rank regression, fatty liver index, type 2 diabetes mellitus, non-alcoholic fatty liver disease

## Abstract

The Fatty Liver Index (FLI) is a proxy for the steatotic component of non-alcoholic fatty liver disease (NAFLD). For sub-Saharan African populations, the contribution of dietary factors to the development of NAFLD in the etiology of type 2 diabetes mellitus (T2DM) remains to be clarified. We identified sex-specific dietary patterns (DPs) related to the FLI using reduced ranked regression (RRR) and evaluated the associations of these DPs with T2DM. This analysis used data from the RODAM, a multi-center cross-sectional study of Ghanaian populations living in Ghana and Europe. The daily intake frequencies of 30 food groups served as the predictor variables, while the FLI was the response variable. The odds ratios and 95% confidence intervals for T2DM were calculated per one standard deviation increase in the DP score using logistic regression. In males, the DP score explained 9.9% of the variation in their food intake and 16.0% of the variation in the FLI. This DP was characterized by high intakes of poultry, whole-grain cereals, coffee and tea, condiments, and potatoes, and the chance of T2DM was 45% higher per 1 DP score-SD (Model 2). Our results indicate that the intake of modernized foods was associated with proxies of NAFLD, possibly underlying the metabolic pathways to developing T2DM.

## 1. Introduction

In sub-Saharan Africa (SSA) and in African migrants living in Europe, type 2 diabetes mellitus (T2DM) constitutes a major health problem [1,2], and the relationships with non-alcoholic fatty liver disease are not fully understood [3]. NAFLD is a spectrum of liver disease primarily characterized by the accumulation of fat in the liver in the absence of excessive alcohol consumption and other chronic liver diseases, such as viral infection [4,5,6]. It is a common liver disorder with an estimated global prevalence of 25% [5,7]. NAFLD strongly co-occurs in patients with components of metabolic syndrome, especially central obesity and T2DM. Indeed, up to 70% of patients with T2DM have NAFLD, and they stand a higher risk of developing non-alcoholic steatohepatitis and advanced fibrosis [6,8,9]. In fact, it is estimated that about 68–74% of individuals with T2DM have evidence of steatosis, and approximately 15% may have advanced fibrosis [9].

To date, the rapid economic transition and urbanization in SSA as well as the sudden lifestyle changes upon migration from Africa to Europe are among the proposed reasons for the upsurge of T2DM among these populations [10]. The parallel nutrition transition comprises shifts in dietary practices from plant-based, fiber-rich and low-fat diets to modernized diets rich in saturated fats and simple carbohydrates. These dietary changes may play a key role in the development of T2DM and NAFLD among SSA populations [11]. Diets high in refined carbohydrates and saturated fats are associated with increased intrahepatic triglycerides, obesity and being overweight, insulin resistance and NAFLD [12,13]. Conversely, the Mediterranean diet, which is characterized by high intakes of fiber-rich foods such as vegetables and whole grains, improves liver function [14,15]. Although dietary interventions are essential in the prevention and primary management of NAFLD, the contributions of dietary factors to the development of NAFLD and the etiology of T2DM remain to be clarified, particularly in SSA populations with a high prevalence of obesity and T2DM [1,2].

Here, we present two methodological advancements to study the etiologic pathways from a diet through NAFLD up to T2DM in SSA populations under various transitions. First, the method of reduced rank regression (RRR) is applied to derive dietary patterns (DPs) related to the proxy markers of NAFLD. RRR determines linear combinations of food groups as the predictor variables by maximizing the explained variation in these proxy markers as the response variables [16]. This approach advances the exploratory nature of principal component analysis and the hypothesis-driven techniques of constructing a priori indices [17,18]. Second, the Fatty Liver Index (FLI) is used as a non-invasive proxy for hepatic steatosis in NAFLD, incorporating both anthropometric and biochemical markers [19,20], and it seems to perform well at any degree of obesity [21].

Considering that NAFLD and T2DM frequently co-occur within subjects and share similar risk factors, we hypothesized that RRR-derived DPs related to NAFLD and characterized by modernized foods are associated with T2DM. Therefore, this study aimed to establish sex-specific DPs related to FLI by RRR and determine the associations between these DPs and T2DM among Ghanaian populations under transition.

## 2. Materials and Methods

### 2.1. Study Design and Population

The Research on Obesity and Diabetes among African Migrants (RODAM) study is a multi-center cross-sectional study that aims to investigate the relative contributions of lifestyle and other risk factors to obesity and T2DM among African migrants in Europe and their counterparts living in rural and urban Ghana. A detailed description of the study design and procedures has been presented elsewhere [22]. In brief, participant recruitment was carried out between July 2012 and September 2015, and it included 6385 Ghanaian adults between the ages of 18 and 96 years living in rural Ghana (*n* = 946), urban Ghana (*n* = 1619), Amsterdam (*n* = 1900), London (*n* = 1258) and Berlin (*n* = 662). For recruitment in Ghana, the census data of 2010 were used to draw urban (Kumasi and Obuasi) and rural participants from the Ashanti Region. The response rates in rural and urban Ghana were 76% and 74%, respectively. In Amsterdam, random selection was performed by using the municipal register to select Ghanaian migrants, who were invited by postal mail and home visits. In Amsterdam, 67% replied by response card or after a home visit, and of these, 53% of the respondents consented to participate in the study. In London, of those individuals who were invited based on their registration in Ghanaian organizations, 75% agreed and participated in the study. Berlin had a participation rate of 68%.

The study protocol was reviewed and approved by the local ethics committees in Ghana (School of Medical Sciences, Komfo Anokye Teaching Hospital, Committee on Human Research, Publication and Ethical Review Board), the Netherlands (Institutional Review Board of AMC, University of Amsterdam), the UK (London School of Hygiene and Tropical Medicine Research Ethics Committee) and Germany (Ethics Committee of Charite-Universitatsmedizin Berlin). All participants who took part in the study were thoroughly informed about the purpose and procedures of the study and gave written informed consent.

### 2.2. Assessment of NAFLD Proxy Markers

Fasting venous blood samples were collected by trained personnel according to standard operating procedures. Serum and plasma samples were obtained through centrifugation and stored at −80 °C until analysis. The enzymatic method (hexokinase) was used to measure the fasting blood glucose in the plasma. All biochemical analyses were performed using an ABX Pentra 400 chemistry analyzer (ABX Pentra; Horiba ABX, Germany). T2DM was defined according to the WHO guidelines as fasting plasma glucose ≥7.0 mmol/L or documented use of glucose-lowering medication or self-reported diabetes. The serum lipid profile (total cholesterol, triglyceride, high-density lipoprotein cholesterol and low-density lipoprotein cholesterol) was measured using immunoturbidimetric assays. The biomarkers of inflammation and liver function comprised C-reactive protein (CRP), aspartate aminotransferase (AST), alanine aminotransferase (ALT) and γ-glutamyl transferase (GGT). We used two sets of proxy markers to define NAFLD. These were the FLI as a single marker and a selection of biomarkers as the second set of proxy markers. The FLI was calculated for each participant according to the algorithm by Bedogni et al. (2006) [23]. The formula considers the BMI, waist circumference, triglycerides and GGT. In the absence of excessive alcohol intake (>21 units (168 g) per week for men and >14 units (112 g) per week for women), the FLI supports the definition of NAFLD in our epidemiological study.

### 2.3. Dietary Assessment

A semi-quantitative Ghana-specific food propensity questionnaire (Ghana-FPQ) was used to determine the usual dietary intake of 134 food items. The usual weekly intake frequencies of food groups were documented in pre-defined portion sizes for the past 12 months. We calculated the amounts of food intake (g/d) using common household measures and translated these into energy intake and nutrient consumption by applying the West African Food Composition Table (2012) and the German Nutrient Database (2010). The food items were grouped into 30 food groups according to their culinary use and nutrient profiles [10]. The food groups subjected to RRR included whole-grain cereals, refined cereals, sweet spreads, dairy products, fruits, nuts and seeds, roots, tubers and plantains, potatoes, fermented maize products, vegetables, legumes, vegetable soups, stews, sauce, rice and pasta, eggs, red meat dishes, cakes and sweets, coffee and tea, alcoholic beverages, sodas and juices, palm oil, olive oil, other oils, margarine, cooking fats and condiments.

### 2.4. Assessment of Covariates

All information on the socio-demographics, education levels, medical histories and lifestyle factors was collected using a structured questionnaire, either self-administered or by an interviewer in the preferred language. The educational status of the participants was categorized into four levels: never or elementary, lower, intermediate and higher or tertiary. Physical activity was assessed using the WHO surveillance questionnaire, and the answers were subsequently classified based on the guidelines of the IPAQ group into three levels of total physical activity: low, moderate and high [24]. Smoking status was classified as current, former or non-smokers. At all study sites, physical examinations were conducted with validated devices according to standard operating procedures and by trained personnel. For all the study participants, anthropometric measurements were taken in light clothes and without shoes. These included weight (kg) by a person scale, height (cm) by a stadiometer, and waist circumference (cm) using a measuring tape (all devices SECA, Germany). We calculated the Body Mass Index (BMI) as the weight over the height squared (kg/m^2^).

## 3. Statistical Analysis

### 3.1. Descriptive Statistics

Data processing and analysis was performed using the Statistical Analysis Software (SAS) (version 9.4). Participants with missing data for the variables of interest were excluded from the present analysis. Details of the exclusions are presented in Appendix A. The final analytical sample comprised 3687 participants. Normally distributed continuous variables are presented as means ± standard deviations (SDs) with the skewed variables as medians (interquartile ranges). All categorical variables are presented as percentages.

### 3.2. Reduced Rank Regression

Owing to the differences in dietary intakes, adipose tissue distribution, and observed T2DM prevalence between males and females, all subsequent analyses were conducted separately for males and females. First, we used RRR to derive dietary pattern (DP) scores related to the FLI as a proxy marker of NAFLD. The RRR method determined the linear combinations of predictor variables (= 30 food groups) that explained as much variation as possible in the response variable (log transformed FLI). This method has been explained in detail by Hoffmann et al. (2004) [25]. Second, we created RRR-derived DP scores using established biomarkers of NAFLD as the response variables (i.e., AST, ALT, GGT, CRP, total cholesterol, HDL-cholesterol, LDL-cholesterol and triglycerides). Hence, four different DPs were derived that explained a maximum of variation in the FLI (for men and women) and in the biomarkers of NAFLD (for men and women).

### 3.3. Association Analyses

The associations between the RRR-derived DPs and T2DM were calculated by logistic regression analyses. The odds ratios and their 95% confidence intervals (CIs) for T2DM were calculated across the quintiles of the RRR-derived DP scores by using the first quintile as the reference category and per 1 score-SD. To assess the linearity assumption, we calculated the odds ratios and their 95% confidence intervals (Cls) across the quintiles of the RRR-derived DP scores, using the first quintile as the reference category. Linear trends across the quintiles were tested by modeling the median of each quintile as a continuous variable. For the associations with the FLI-related DP scores, we sequentially adjusted for the age (years) and study site (categorical) (Model 1) as well as the education level (4 categories), energy intake (kcal/d), smoking status (yes or no), physical activity (METs-h/week), and alcohol consumption (g/day) (Model 2). For the associations with the DPs related to the biomarkers, we additionally adjusted for the BMI and waist circumference (Model 3).

### 3.4. Sensitivity Analysis

In sensitivity analyses, we excluded participants with self-reported T2DM to address a potential source of reverse causation. Specifically, individuals with known T2DM who received medical treatment might have presented altered biomarkers and could have changed their diets. Consequently, we repeated all RRR analyses using the FLI or NAFLD biomarkers as response variables for both males and females. In addition, we repeated the final regression analysis for the association between RRR-derived DPs and T2DM, applying an HbA1c-based definition of the outcome, namely HbA1c ≥ 48 mmol/mol, self-reported diabetes or the use of glucose-lowering medication.

## 4. Results

### 4.1. Study Population

Table 1 shows the socio-demographic, anthropometric, lifestyle, and clinical characteristics of the RODAM study population by study site and sex. The majority of the participants were females (63.0%) and middle-aged (46.1 ± 11.1 years). The males were older than the females, but the mean age was similar across the study sites (Table 1). Males had higher attained educations than females, and the proportion of participants with higher education was highest in London. Males were more physically active, more likely to be former or current smokers and reported higher alcohol consumption than females. The mean BMI (24.8 ± 4.4 kg/m^2^ vs. 27.8 ± 5.7 kg/m^2^) and waist circumference (87 ± 12.1 cm vs. 91 ± 12.5 cm) were lower in males than in females. The participants in London showed the lowest level of physical activity (median: 28; IQR: 5–112 min/day) and the highest BMI (29.4 ± 4.8 kg/m^2^) and waist circumference (95.4 ± 11.3cm), while rural Ghana showed the lowest BMI (22.7 ± 4.3 kg/m^2^) and waist circumference (81.2 ± 10.9 cm) and the highest physical activity level (median: 90; IQR: 36–161 min/day).

### 4.2. Intakes of Energy, Nutrients and Food Groups

The estimated mean energy intake was 2533 ± 837 kcal/d, and this was higher among males than females and highest in Berlin, followed by London, rural Ghana, Amsterdam and urban Ghana (Table 1). The mean contributions to the daily energy intake were 53.0 ± 9.1 energy% for carbohydrates, 32.2 ± 8.2 energy% for total fats and 13.8 ± 2.9% for protein. This was similar between males and females. The contribution of macronutrients to energy intake varied across study sites. In rural and urban Ghana, carbohydrates were the major contribution to the energy percentage, while in London and Amsterdam, fats and proteins contributed the most to energy intake. The consumption of certain foods like coffee and tea, sodas and juice and alcohol was higher in males compared with females (Figure 1a). This was also seen in red meat, poultry, processed meat and eggs, while women consumed more vegetarian mixed dishes, cakes and sweets than men (Figure 1b).

### 4.3. RRR-Derived Dietary Patterns Related to the Fatty Liver Index

Table 2 shows the explained variation and the factor loadings of those food groups related to the FLI as derived by RRR for males and females. The DP scores explained 9.9% and 6.5% of the total variation in food group intakes in males and females, respectively. Furthermore, the RRR-derived pattern scores were positively related with the FLI and explained 16.0% of the FLI variation among males (beta = 0.4) and 8.8% of the FLI variation among females (beta = 0.3). The DPs scores in males were characterized by high intakes of poultry, whole-grain cereals, coffee and tea, condiments and potatoes, and they were inversely related with palm oil, roots, tubers and plantains, refined cereals and fermented maize products. Among females, the DPs scores were characterized by frequent consumption of coffee and tea, poultry, whole-grain cereals, margarine, fish and alcoholic beverages and by low intakes of roots, tubers and plantains, fermented maize products, palm oil and refined cereals.

In the sensitivity analysis (Appendix A), similar DP scores were obtained when self-reported T2DM cases were excluded. Among males, the explained variation in food group intake was 9.6%, and the explained variation in the FLI was 15.5%. In females, these figures were 6.3% and 8.9%, respectively.

### 4.4. Associations of FLI-Related Pattern Scores with T2DM

The associations with T2DM are shown per quintile and per one SD of the RRR-derived DP scores in Table 3. Among males, a higher adherence to the DP score was associated with increased odds of T2DM. As a trend, this was seen across the DP score quintiles, and these associations were significant per one score-SD increase. In the crude model, the odds of T2DM were 55% higher per 1 score-SD (95% CI: 1.30–1.86). The association attenuated after adjustment for the age and study site (Model 1: 1.34; 95% CI: 1.04–1.73) and remained stable after further adjustment for socioeconomic and lifestyle factors (Model 2: 1.45; 95% CI: 1.10–1.93). Among females, the associations were weaker. The RRR-derived DP score was positively associated with T2DM per one score-SD increase (crude model: 1.24; 95% CI: 1.07–1.44) but attenuated after adjustment for demographic, socioeconomic and lifestyle factors (Model 2: 1.16; 95% CI: 0.95–1.42).

In the sensitivity analysis (Appendix A), we applied an HbA1c-based definition of T2DM as an outcome variable. This produced a stronger association between the RRR-derived DP scores and T2DM for both men and women. In the final model, the associations were OR: 1.54 (95% CI: 1.19–1.99) for men and OR: 1.27 (95% CI: 1.07–1.52) for women (Appendix A).

### 4.5. RRR-Derived Dietary Patterns Related to the NAFLD Biomarkers

In addition to the FLI-related DP scores, we also extracted two DP scores related to the biomarkers of NAFLD (i.e., liver enzymes, blood lipids and CRP). Table 4 shows these DP scores and the corresponding factor loadings of the food groups. Among males, the DP score explained 11.2% of the total variation in the food group intake, and it was characterized by high intakes of whole-grain cereals, poultry, dairy products, coffee and tea, condiments, potatoes, margarine and olive oil. Again, this DP correlated with rare intakes of palm oil, roots, tubers and plantains and fermented maize products. In women, the RRR-derived DP score explained 12.1% of the total variation of the food intake, and it was correlated with frequent intakes of palm oil, vegetarian mixed dishes and fish. Interestingly, the DP score among females correlated inversely with food groups that loaded positively among females, namely whole-grain cereals, poultry, dairy products, coffee and tea, condiments, potatoes, margarine and olive oil.

The explained variations in the NAFLD biomarkers and the response weights for the RRR-derived DP scores are presented in Table 5. Among males, the DP score explained 2.8% of the total biomarker variation, with the major contribution to AST variation (7.4%) and a response weight of −0.58. This was followed by total cholesterol (4.5%) and LDL-cholesterol (4.2%), both showing positive response weights with the RRR-derived DP score. The relationships with AST were mainly driven by whole-grain cereals and coffee and tea (both r = −0.20). In addition, the positive response weights for the total cholesterol and LDL-cholesterol were mainly attributed to whole-grain cereals, potatoes and condiments (Appendix A). In females, the explained total variation in the NAFLD biomarkers was 4.5%, with major contributions to AST at 12.3% (response weight: 0.58), triglycerides at 11.1% (response weight: 0.55) and HDL-cholesterol at 8.6% (response weight: −0.49) (Table 5). These relationships for AST and triglycerides were mainly driven by vegetarian mixed dishes and palm oil (both r = 0.20), and that for HDL-cholesterol was driven by coffee and tea and potatoes (Appendix A). In our sensitivity analysis, after exclusion of participants with self-reported T2DM, we observed similar DP scores related to the NAFLD proxy markers as in the full dataset. Among males, the DP scores explained 11.0% of the total variation in the food group intake and 3.3% of the total variation in the biomarkers of NAFLD (Appendix A). Among females, the DP scores explained 12.0% of the variation in food groups and 4.5% of the total variation in the biomarkers of NAFLD (Appendix A).

### 4.6. Associations of NAFLD Biomarker-Related Pattern Scores with T2DM

Table 6 shows the odds of T2DM across quintiles and per one SD of the RRR-derived DP scores related to the NAFLD biomarkers, stratified by sex. Among males, the crude model showed a positive association with T2DM status (*p* for trend across quintiles = 0.0002; OR per 1 score-SD increase: 1.42; 95% CI: 1.20–1.68). However, this association attenuated to the null when adjusting for demographics, socioeconomics, lifestyle and anthropometry. Contrary to the males, there was no clear linear trend for the associations across quintiles with T2DM status (*p* for trend >0.05). Positive associations ranged from 8% increased odds of T2DM in the crude model (95% CI: 0.93–1.25) to 30% increased odds of T2DM in Model 3 (95% CI: 0.99–1.71) per one score-SD increase.

Again, we conducted sensitivity analysis using an HbA1c definition as the outcome variable (Appendix A). The corresponding associations of the biomarker-related DP scores were weaker than those in the primary analysis. In the final Model 2 results, the associations were OR: 1.01 (95% Cl: 0.78–1.32) among men and OR: 1.27 (95% CI: 0.99–1.62) among women (Appendix A).

## 5. Discussion

The relationships between NAFLD and T2DM with respect to their dietary risk factors informed the derivation of DP scores by applying the RRR method. We used two sets of proxy markers for NAFLD (1: FLI; 2: serum liver enzymes, blood lipids and CRP) as response variables and the habitual consumption of 30 food groups as predictor variables. The FLI-related DPs were characterized by frequent intakes of modernized foods and were positively associated with the odds of T2DM. This association was stronger in males than in females. The RRR-derived DP scores using CRP, liver enzymes and blood lipids as the response variables showed inconsistent relationships with food intake between males and females and implausible associations with T2DM.

### 5.1. Dietary Patterns and Proxy Markers of NALFD and T2DM

In our first approach, the FLI was used as a proxy marker for NALFD and as the response variable in the RRR analysis. The respective FLI-related DPs were similar between males and females and were characterized by frequent intakes of poultry, whole-grain cereals, coffee and tea, condiments, potatoes, alcoholic beverages, margarine and fish, while fermented maize products, refines cereals, roots, tubers and plantains and palm oil showed inverse correlations with the FLI. These DPs correspond to the nutrition transition observed for SSA populations when experiencing rapid economic growth, accelerated urbanization and migration to Europe [10,26].

Our findings partly accord with recent evidence from a meta-analysis on dietary risk factors of NAFLD [27]. Red meat and sugar-sweetened beverages are robustly associated with a higher risk of NAFLD, while nuts are consistently associated with lower NAFLD risk. This meta-analysis demonstrates that many foods have neutral relationships with NAFLD, such as dairy, fish, rice, noodles and whole-grain products [27]. In our study, we also identified positive factor loadings for condiments, meat (poultry) and coffee and tea, which showed positive correlations with the FLI. For coffee and tea, this appears to contradict the observed lower risk of elevated liver enzymes associated with frequent coffee and tea consumption [28]. The antioxidant properties of these beverages may exhibit a suppressive effect of hyperglycemia by enhancing insulin sensitivity and subsequently improving a fatty liver [29]. However, coffee and tea are often sweetened with sugar and condensed milk, which happens to be underreported during dietary assessments. Thus, the high glycemic load of sweetened beverages and condiments may explain the positive correlations with the FLI in our study. Reassuringly, in a large Chinese population, an RRR-derived DP related to inflammatory markers is also characterized by high intakes of sugar-containing foods and positively associated with NAFLD [30]. In fact, low glycemic index diets are currently promoted for the amelioration of NAFLD [31].

While many studies confirm the benefits of poultry and fish as part of a healthy diet [32,33], this may not apply uniformly. Review data and recent findings from the multi-ethnic cohort show that poultry intake can be associated with higher risk of NAFLD [27,34]. The underlying mechanisms remain to be investigated, as most studies did not examine poultry separately but focused on overall meat consumption. Additionally, dietary patterns rich in fish intake are known to prevent NAFLD, mainly due to the beneficial effects of omega-3 poly-unsaturated fatty acids on health [33]. Yet, the increased contamination of sea fish with environmental toxins might cause harmful effects to the human endocrine system and promote NAFLD [35], which could contribute to our findings. In light of this, our study reaffirms the current dietary recommendations for the prevention of NAFLD and T2DM, including foods with low glycemic indexes that are rich in poly-unsaturated fatty acids [27].

Interestingly, we observed stronger relationships for the FLI-related DPs among males compared with females, despite the fact that the characteristic food groups and the explained variation in the FLI were similar between sexes. This may reflect the sex-specific contributions of NAFLD to the etiology of T2DM. Indeed, recent findings from the RODAM study confirmed the increased odds of T2DM among participants with an FLI ≥60 compared with an FLI <60. This association is stronger in men (OR: 2.43; 95% CI: 1.73–3.41) than in women (OR: 2.02; 95% CI: 1.52–2.69) [36]. Further studies found a positive association between FLI and metabolic-associated fatty liver diseases (MAFLD) [36]. This observation was attributed to several factors, including diet-related changes in the composition of the gut microbiota, which in turn may influence the development of MAFLD. Male adults from Ghana and Kenya present a lower prevalence of an increased FLI and higher prevalence of T2DM than females [37,38]. The explanation for the observed sex-specific relationships between the FLI—as a proxy marker of NAFLD—and T2DM may lay in the FLI components. It is well-established that measures of central adiposity, such as the waist circumference and waist-to-hip ratio, predict the risk of cardiometabolic diseases among SSA populations better than the measure of general obesity (BMI), particularly among females [39,40]. The fact that females of African ancestry tend to have more subcutaneous and less visceral fat at any given level of the BMI than men supports the notion that female individuals can metabolically tolerate a certain degree of obesity as opposed to males [41]. In addition, the biomarker components of the FLI may contribute to the observed sex differences, including serum triglycerides and HDL-cholesterol [37].

Additionally, in an urban Ghanaian case control study, Frank et al. derived a DP using RRR that has been positively related with serum triglycerides and inversely related with adiponectin. This pattern is associated with higher odds of T2DM and shares similar features with the RRR-derived DPs in the present analysis, being characterized by high consumption of starchy foods and low intake of fruits and vegetables [16]. The combination of condiments, margarine and higher glycemic index foods such potatoes may contribute to the observed associations with T2DM [42,43]. These foods characterize modernized diets, which are known to confer an unhealthy body composition and increased risk of T2DM [44]. In addition, our findings are partly corroborated by recent evidence from a meta-analysis of 16 cohort studies. In this study, the RRR-derived DPs were characterized by high intakes of refined grains, sugar-sweetened soft drinks and processed meat and were significantly associated with T2DM risk [45]. In our study, the DPs were also characterized by sugary foods such as condiments and sweetened hot beverages and were positively associated with T2DM. Notably, previous exploratory approaches in this Ghanaian study population identified three DPs using PCA that had either no or a negative association with T2DM [46]. A rice, pasta, meat and fish DP showed increased odds of T2DM by one SD-score increase but overlapped with the current RRR-derived DPs only regarding condiments and fish. The strongest similarities of the RRR-derived DPs were seen for the previously identified mixed DP, which was not associated with T2DM. At the same time, the strongest dissimilarities were observed for the roots, tubers and plantains DP, which tended to be positively associated with T2DM. These findings may reflect the multiple pathways through which dietary factors contribute to the etiology of T2DM, with NAFLD being only one possible link.

In our second approach, we used a selection of proxy markers for NAFLD as response variables for the RRR analysis, comprising liver enzymes, blood lipids and CRP. In this analysis, we observed inconsistent DP scores between men and women and biologically implausible associations with T2DM. Presumably, this selection of biomarkers may not be specific enough to operationalize NAFLD, as they also reflect other metabolic pathways to T2DM, including chronic inflammation (CRP) and dyslipidemia (blood lipids) [47], which may explain the unexpected results in this analysis. In addition, unmeasured confounding might have distorted our findings with regard to biomarker-related DPs.

### 5.2. Strengths and Limitations

When interpreting the findings of our study, some limitations need to be acknowledged. First, the definition of NAFLD by the use of the FLI as a proxy marker without liver biopsy or imaging may shield detailed information on the severity classification of a fatty liver [48]. However, liver biopsy is invasive, associated with discomfort and often technically unfeasible in large-scale epidemiological studies. In contrast, the FLI has proven to be a useful predictor for a fatty liver, since it shows high concordance with the imaging and histological criteria for NAFLD [49]. At the same time, we acknowledge that validation of the FLI among sub-Saharan African populations is still pending [36]. As data on viral hepatitis status was not available in our study, misclassification of NAFLD cannot not fully be excluded. However, the study provides unique insights into diet-disease pathways for T2DM in a large sample of adults from sub-Saharan Africa. The method for DP construction accounts for the potential links between NAFLD and T2DM through projecting the response variable of the FLI on the food groups as predictor variables. We are cognizant that this approach has been criticized for its choice of proxy measures, which could simply reflect the early disease stages of T2DM [50]. In fact, previous data from Ghanaian adults suggest that increased serum triglycerides serve as an important risk factor for T2DM [16,51], while the BMI does not [1,39]. For the associations of FLI-related DP scores with T2DM, we did not adjust for anthropometric components of the FLI in our final regression model, which might have caused some residual confounding and overestimation of the observed DP–T2DM relationships, as the FLI formula includes the BMI and waist circumference. Like in all cross-sectional studies, reverse causation cannot be ruled out in our study either, particularly for the bidirectional relationship between NAFLD and T2DM, given the fact that some study participants with T2DM had received treatment, including lifestyle counseling. Still, the DP scores remained when individuals with self-reported T2DM were excluded. Finally, we have addressed potential confounding by our comprehensive sets of adjustment variables. However, unmeasured confounding, such as a family history of diabetes, cannot be ruled out.

## 6. Conclusions

We identified DPs that showed adherence to modernized foods and a positive relation with the FLI, a proxy marker for fatty liver disease. These DPs were positively associated with the odds of T2DM, particularly in males. These findings give credence to the concept that modernized dietary practices among Ghanaian populations under transition may underly metabolic pathways to NAFLD and T2DM.

## Figures and Tables

**Figure 1 nutrients-13-03679-f001:**
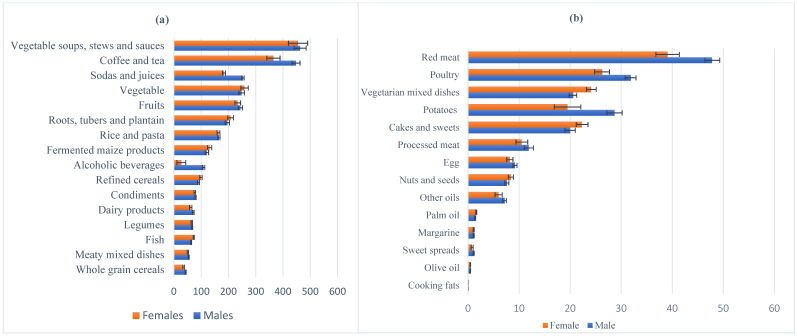
Mean intakes and standard deviations (g/day) of (**a**) 16 food groups with a mean intake of ˃50 g/day and (**b**) 14 food groups with a mean intake of ≤50 g/day.

**Table 1 nutrients-13-03679-t001:** Characteristics and biomarkers of the RODAM study population by sex and site ^1^.

Characteristics	Total (*n* = 3687)	Men (*n* = 1366)	Women (*n* = 2321)	Rural Ghana (*n* = 820)	Urban Ghana (*n* = 1358)	Amsterdam (*n* = 707)	Berlin (*n* = 451)	London (*n* = 351)
Sex (female%)	63.0	-	-	61.6	72.2	60.1	45.0	59.0
Age (years)	46.1 ± 11.1	46.9 ± 11.3	45.6 ± 10.9	46.7 ± 12.6	45.3 ± 11.4	46.6 ± 8.5	45.2 ± 10.4	47.9 ± 10.9
Education								
Never or elementary%	37.6	21.9	46.9	56.8	43.9	35.5	9.3	9.1
Low	37.7	41.6	35.5	31.6	38.9	37.8	50.1	31.6
Intermediate	16.2	22.5	12.5	7.9	12.5	21.8	26.6	24.8
Higher vocational	8.5	14.1	5.2	3.7	4.71	5.0	14.0	34.5
Length of stay (years)	-	-	-	-	-	16.4 ± 8.1	17.0 ± 10.9	17.2 ± 11.0
Body mass index (kg/m^2^)	26.7 ± 5.5	24.8 ± 4.4	27.8 ± 5.7	22.7 ± 4.3	26.9 ± 5.4	28.9 ± 5	27.6 ± 4.8	29.4 ± 4.8
Waist circumference (cm)	89.5 ± 12.5	87 ± 12.1	91 ± 12.5	81.2 ± 10.9	89.4 ± 11.8	94.6 ± 11.6	92.2 ± 11.5	95.4 ± 11.3
Smoking (current or former%)	9.3	19.6	3.2	7.9	6.9	11.6	18.4	5.4
Physical activities (MET-min/day)	72 (14–168)	96 (28–196)	56 (10–156)	90 (36–161)	60 (6–156)	88.7 (26–258)	72 (12–198)	28 (5–112)
Total energy intake (kcal/day)	2533 ± 837	2628 ± 827	2477 ± 817	2594 ± 828	2298 ± 661	2478 ± 854	2929 ± 944	2898 ± 953
Carbohydrate intake (energy%)	53.0 ± 9.1	52.2 ± 9.5	53.5 ± 8.9	56.4.5 ± 8.3	54.5 ± 8.1	50.5 ± 8.3	48.5 ± 10.9	50.2 ± 9.6
Fat intake (energy%)	32.2 ± 8.2	32 ± 8.6	32.3 ± 8	31.4 ± 7.3	31.6 ± 7.2	32.1 ± 8.3	33.7 ± 10.6	34.1 ± 9.6
Protein intake (energy%)	13.8 ± 2.9	13.9 ± 3.1	13.8 ± 2.9	11.5 ± 2.2	13.6 ± 2.4	15.8 ± 2.7	14.8 ± 3.1	15.1 ± 2.9
Alcohol intake (g/day)	0 (0–0.1)	0 (0–0.3)	0 (0–0.1)	0 (0–0.1)	0 (0–0.1)	0.1 (0–0.4)	0.1 (0–0.6)	0 (0–0.1)
AST U/L	32.3 (26.6–39.8)	35.1 (29.1–43.1)	30.6 (25.3–37.7)	36.1 (30.4–43.1)	34.4 (28.7–41.5)	26.1 (22.4–30.8)	28.9 (24.7–34.9)	34.1 (28.0–43.1)
ALT U/L	19.2 (14.9–25.7)	23.0 (17.4–31.2)	17.6 (13.9–22.7)	19.2 (15–24.9)	19.3 (15–25.8)	17.4 (13.7–23.0)	19.9 (14.8–26.9)	22.5 (18.3–30.3)
GGT U/L	30.8 (23.2–43.1)	37.4 (27.3–52.6)	27.9 (21.7–37.3)	29.5 (22.3–42.2)	31.4 (23.9–42.9)	30.2 (22.8–42.0)	32.9 (24.7–46.1)	30.6 (22.7–43.9)
CRP mg/L	0.7 (0.2–2.5)	0.5 (0.1–1.5)	0.9 (0.2- 3.2)	0.7 (0.2–2.6)	0.9 (0.2–3.1)	0.8 (0.2–2.3)	0.5 (0.2–1.9)	0.8 (0.2–2.3)
Total cholesterol (mmol/L)	5.0 ± 1.1	4.9 ± 1.1	5.1 ± 1.1	4.6 ± 1.1	5.2 ± 1.2	5.0 ± 1.1	5.1 ± 1.1	5.0 ± 1.0
LDL-cholesterol (mmol/L)	3.2 ± 1.0	3.1 ± 1.0	3.2 ± 1.0	2.8 ± 1.0	3.4 ± 1.0	3.2 ± 0.9	3.2 ± 1.0	3.3 ± 0.9
HDL-cholesterol (mmol/L)	1.3 ± 0.4	1.3 ± 0.4	1.3 ± 0.4	1.2 ± 0.4	1.3 ± 0.3	1.4 ± 0.3	1.5 ± 0.4	1.3 ± 0.3
Triglycerides (mmol/L)	0.9 (0.7–1.2)	1.0 (0.7–1.3)	0.9 (0.7–1.2)	1.0 (0.8- 1.3)	1.0 (0.8–1.3)	0.8 (0.6–1.0)	0.9 (0.6–1.1)	0.8 (0.6–1.1)
Fatty Liver Index	2.6 ± 6.3	2.0 ± 5.4	2.9 ± 6.8	1.0 ± 3.2	2.8 ± 6.7	3.3 ± 7.2	3.1 ± 7.3	3.1 ± 6.0

^1^ Continuous variables are expressed as means ± SDs or medians (IQR). Categorical variables are expressed as percentages (%). AST: aspartate aminotransferase; ALT: alanine aminotransferase; CRP: C-reactive protein; GGT: gamma-glutamyl transferase; HDL: high-density lipoprotein; LDL: low-density lipoprotein.

**Table 2 nutrients-13-03679-t002:** Percentage of explained variation in food intake and factor loadings of 30 food groups of the RRR-derived dietary pattern scores related to the FLI for males and females ^1^.

Food Group	Men (*n* = 1366)		Women (*n* = 2321)	
	Explained Variation (%)	Factor Loading	Explained Variation (%)	Factor Loading
Poultry	**30.8**	**0.32**	**16.6**	**0.29**
Whole-grain cereals	**21.5**	**0.27**	**15.8**	**0.29**
Coffee and tea	**19.9**	**0.26**	**21.0**	**0.33**
Condiments	**18.3**	**0.25**	**15.7**	**0.28**
Potatoes	**17.8**	**0.25**	2.1	0.10
Alcoholic beverages	**10.3**	**0.19**	**7.5**	**0.20**
Margarine	**10.2**	**0.19**	**10.6**	**0.23**
Olive oil	7.7	0.16	0.4	0.04
Processed meat	7.2	0.16	0.3	0.04
Other oils	5.5	0.14	2.4	0.11
Dairy products	4.9	0.13	0.4	0.04
Sodas and juices	3.8	0.11	1.8	0.10
Cakes and sweets	3.4	0.11	0.6	−0.05
Red meat	2.6	0.09	0.0	0.01
Vegetables	2.5	0.09	5.2	0.16
Sweet spreads	1.7	0.08	0.0	−0.01
Cooking fats	1.5	0.07	0.2	−0.03
Eggs	1.3	0.07	2.1	−0.10
Rice and pasta	1.1	0.06	1.7	0.09
Vegetable soups, stews and sauces	0.9	0.06	0.0	−0.01
Nuts and seeds	0.5	0.04	1.5	0.09
Fish	0.1	0.02	**9.5**	**0.22**
Meat mixed dishes	2.1	−0.08	1.3	−0.08
Fruits	4.1	−0.12	1.6	−0.09
Legumes	4.8	−0.13	2.4	−0.11
Vegetarian mixed dishes	7.2	−0.16	0.9	−0.07
Fermented maize products	**18.4**	**−0.25**	**19.8**	**−0.32**
Refined cereal	**22.4**	**−0.27**	**13.3**	**−0.26**
Roots, tubers and plantains	**28.7**	**−0.31**	**22.0**	**−0.34**
Palm oil	**35.1**	**−0.34**	**17.3**	**−0.30**
**Total**	**9.9**		**6.5**	

^1^ Factor loadings are correlations between food groups and the dietary pattern scores. Figures in bold represent food items with relevant contributions to the dietary pattern score (≥0.20% explained variation in the factor loadings for either males or females).

**Table 3 nutrients-13-03679-t003:** Associations of the FLI-related and RRR-derived dietary pattern scores with type 2 diabetes among males and females ^1^.

Model	Odds Ratio (95% Confidence Interval)						
	Q1	Q2	Q3	Q4	Q5	*p* for Trend	Per 1 Score-SD
**Men**							
Diabetes/Control	17/256	22/251	31/243	46/227	43/230		
Crude	1 (reference)	1.32 (0.69, 2.54)	1.92 (1.06, 3.56)	3.05 (1.70, 5.47)	2.82 (1.56, 5.07)	<0.0001	1.55 (1.30, 1.86)
Model 1	1 (reference)	1.41 (0.71, 2.79)	1.79 (0.90, 3.56)	2.30 (1.07, 4.97)	1.97 (0.89, 4.35)	0.11	1.34 (1.04, 1.73)
Model 2	1(reference)	1.25 (0.62, 2.49)	1.58 (0.79, 3.16)	2.14 (0.98, 4.68)	2.03 (0.90, 4.60)	0.07	1.45 (1.10, 1.93)
**Women**							
Diabetes/Control	25/439	38/426	32/433	47/417	47/417		
Crude	1(reference)	1.57 (0.93, 2.64)	1.30 (0.76, 2.23)	1.98 (1.20, 3.27)	1.98 (1.20, 3.27)	<0.005	1.24 (1.07, 1.44)
Model 1	1(reference)	1.59 (0.93, 2.71)	1.31 (0.75, 2.30)	2.05 (1.19, 3.54)	1.98 (1.09, 3.59)	<0.02	1.23 (1.03, 1.48)
Model 2	1(reference)	1.28 (0.74, 2.21)	1.03 (0.58, 1.83)	1.64 (0.94, 2.84)	1.65 (0.90, 3.02)	0.07	1.16 (0.95, 1.42)

^1^ Odds ratios (ORs) at 95% confidence intervals (CIs) were calculated by logistic regression, and the *p*-values for the trend were calculated by modeling the median of the dietary pattern scores as the independent variable. Model 1: adjusted for age (years) and study site (categorical); Model 2: Model 1 + education (4 categories), energy intake (kcal/day), smoking (yes or no), physical activity (METs-h/week) and alcohol (alcohol/day).

**Table 4 nutrients-13-03679-t004:** Percentage of explained variation in food intake and factor loadings of 30 food groups of the RRR-derived dietary pattern scores related to NAFLD biomarkers among males and females ^1^.

Food Group	Men (*n* = 1366)		Women (*n* = 2321)	
	Explained Variation (%)	Factor Loading	Explained Variation (%)	Factor Loading
Whole-grain cereals	**36.5**	**0.33**	**25.3**	**−0.26**
Poultry	**26.6**	**0.28**	**35.1**	**−0.31**
Dairy products	**25.8**	**0.28**	13.4	−0.19
Coffee and tea	**23.8**	**0.27**	**44.9**	**−0.35**
Condiments	**21.4**	**0.25**	**33.0**	**−0.30**
Potatoes	**19.0**	**0.24**	**28.7**	**−0.28**
Margarine	**12.8**	**0.20**	**14.6**	**−0.20**
Olive oil	**13.6**	**0.20**	**18.3**	**−0.22**
Sodas and juices	7.9	0.15	6.1	−0.13
Sweet spreads	7.8	0.15	6.6	−0.13
Rice and pasta	7.5	0.15	0.7	−0.05
Processed meat	5.5	0.13	5.7	−0.13
Palm oil	**32.7**	**−0.31**	**27.1**	**0.27**
Roots, tubers and plantains	**30.6**	**−0.30**	12.4	0.18
Fermented maize products	**23.5**	**−0.26**	6.1	0.13
Vegetarian mixed dishes	10.5	−0.18	**27.3**	**0.27**
Refined cereals	5.6	−0.13	7.4	0.14
Cakes and sweets	4.0	0.11	7.5	−0.14
Vegetables	3.9	0.11	9.2	−0.16
Meaty mixed dishes	4.1	−0.11	0.9	0.05
Legumes	3.9	−0.11	0.5	−0.04
Other oils	2.8	0.09	4.2	−0.11
Cooking fats	1.5	0.07	0.1	−0.01
Fish	1.6	−0.07	13.1	0.19
Fruits	1.4	−0.06	1.7	−0.07
Eggs	0.9	0.05	5.9	−0.13
Vegetable soups, stews and sauces	0.3	0.03	0.1	0.01
Red meat	0.3	−0.03	1.1	−0.05
Nuts and seeds	0.4	−0.03	0.0	−0.01
Alcoholic beverages	0.1	−0.02	6.3	−0.13
**Total**	**11.2**		**12.1**	

^1^ Factor loadings are correlations between food groups and the dietary pattern score. Figures in bold represent food groups with relevant contributions to the dietary pattern score (≥0.20% explained variation in the factor loading for either males or females).

**Table 5 nutrients-13-03679-t005:** Percentage of explained variation in the NALFD biomarkers and response weights of the RRR-derived dietary pattern scores among males and females ^1^.

Biomarker	Men (*n* = 1366)		Women (*n* = 2321)	
	Explained Variation (%)	Response Weight	Explained Variation (%)	Response Weight
Cholesterol	**4.5**	**0.45**	0.3	0.09
LDL-cholesterol	**4.2**	**0.44**	1.0	0.16
HDL-cholesterol	2.9	0.36	**8.6**	**−0.49**
AST	**7.4**	**−0.58**	**12.3**	**0.58**
GGT	1.5	−0.26	0.3	0.09
Triglycerides	1.3	−0.24	**11.1**	**0.55**
C-reactive protein	0.3	−0.12	0.4	0.11
ALT	0.2	0.08	2.2	0.25
**Total**	**2.80**		**4.50**	

^1^ Figures in bold represent biomarkers with relevant relationships with the dietary pattern scores (response weight >|0.35|).

**Table 6 nutrients-13-03679-t006:** Associations of the biomarker-related and RRR-derived dietary pattern scores with type 2 diabetes among males and females ^1^.

			Odds Ratio (95% Confidence Interval)				
	Q1	Q2	Q3	Q4	Q5	*p* for Trend	Per 1 Score-SD
**Men**							
Diabetes/Control	15/258	27/246	33/241	47/226	37/236		
Crude	1 (reference)	1.89 (0.98, 3.63)	2.36 (1.25, 4.44)	3.58 (1.95, 6.57)	2.7 0(1.44, 5.04)	<0.0002	1.42 (1.20, 1.68)
Model 1	1 (reference)	1.91 (0.97, 3.76)	2.06 (1.00, 4.22)	2.42 (1.07, 5.50)	1.64 (0.72, 3.75)	0.578	1.15 (0.89, 1.48)
Model 2	1 (reference)	1.76 (0.89, 3.50)	1.8 (0.86, 3.75)	2.07 (0.90, 4.78)	1.44 (0.61, 3.40)	0.743	1.13 (0.86, 1.49)
Model 3	1 (reference)	1.62 (0.81–3.22)	1.46 (0.69–3.10)	1.70 (0.72–4.01)	1.21 (0.50, 2.89)	0.962	1.07 (0.81, 1.42)
**Women**							
Diabetes/Control	37/427	38/426	36/429	36/430	44/420		
Crude	1 (reference)	1.03 (0.64, 1.65)	0.97 (0.60, 1.56)	0.91 (0.56, 1.48)	1.21 (0.77, 1.91)	0.592	1.08 (0.93, 1.25)
Model 1	1 (reference)	1.24 (0.76, 2.05)	1.63 (0.84, 3.14)	1.46 (0.73, 2.92)	1.69 (0.84, 3.39)	0.229	1.35 (1.05, 1.73)
Model 2	1 (reference)	1.16 (0.70, 1.94)	1.39 (0.69, 2.79)	1.20 (0.57, 2.52)	1.40 (0.67, 2.93)	0.522	1.29 (0.99, 1.68)
Model 3	1 (reference)	1.11 (0.66, 1.86)	1.32 (0.65, 2.67)	1.20 (0.57, 2.57)	1.42 (0.67, 3.02)	0.422	1.30 (0.99, 1.71)

^1^ Odds ratios (ORs) at 95% confidence intervals (CIs) were calculated by logistic regression, and the *p*-values for the trend were calculated by modeling the median of the dietary pattern scores as the independent variable, Q1 as quintile 1, etc. Model 1: adjusted for age (years) and study site (categorical); Model 2: Model 1 + education (4 categories), energy intake (kcal/day), smoking (yes or no), physical activity (METs-h/week); Model 3: Model 2+ BMI and waist circumference.

## Data Availability

Data will be available upon request from the corresponding author.

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
