# Peer review of "Reduced Rank Regression-Derived Dietary Patterns Related to the Fatty Liver Index and Associations with Type 2 Diabetes Mellitus among Ghanaian Populations under Transition: The RODAM Study"

_nutrients, 2021, doi:10.3390/nu13113679_

Round 1

Reviewer 1 Report

The present study highlights an important point in NAFLD research where diet is thought to be linked with fatty liver disease. I personally think the article is well written and adds clinical value to the study of NAFLD disease. However, there are a few points that I wish to clarify with the authors.

First, the authors use the FLI to measure NAFLD. there are many caveats in that measure without the presence of more robust measure (e.g. biopsy or imaging based ultrasound). The limitations of FLI and lack of biopsy proven fatty liver disease should be highlighted in the manuscript. 

There should also be a proper definition of NAFLD in the methods, i.e. fatty liver in the absence of alcohol use. 

Additionally, why was BMI left out in the variables used for adjustments? BMI is a major cofounder in all NAFLD research and should be included.

Lastly, You may find the following references important to include:

  1. https://dom-pubs.onlinelibrary.wiley.com/doi/abs/10.1111/dom.14521
  2. https://www.gastro.theclinics.com/article/S0889-8553(19)30072-X/abstract
  3. https://www.nature.com/articles/s41575-021-00472-y

Reviewer 2 Report

In the present manuscript the authors report the results of a cross-sectional study investigating the association between dietary patterns and the prevalence of fatty liver disease and type 2 diabetes among Ghanaian individuals residing in Ghana and in Europe. The topic is certainly of interest as it focuses on a very common condition in a specific population. The manuscript is generally well written and clear.

I have the following comments:

  • The major limitation of the present study is the use of the Fatty liver index to identify participants with steatosis. Since this biomarker was derived in an Italian population comprising Caucasian individuals, one could argue that its performance in Ghahanian participants has not been studied so far. Are there studies in the literature that evaluated the performance of FLI in African patients? Otherwise this should be considered a limitation of the study.
  • Diagnosis of diabetes: why did you rely only on fasting plasma glucose? Do wou have data on Hba1c as well? If not, this should be added to the limitation section.
  • Similarly, a single elevated FPG value is not enough to diagnose patients with diabetes. This should be considered as a limitation as well.
  • When discussing the prevalence of NAFLD in patients with diabetes and in the general population I would refer to recent studies investigating not only the prevalence of steatosis, but most importantly that of advanced liver fibrosis (see DOI: 10.1111/liv.14828 and doi:10.2337/dc20-1778)
  • Did you screen for viral hepatitis and other causes of chronic liver disease? This in my opinion is an important point. Secondly, analyses should also be performed after exclusion of participants with significant alcohol consumption.
  • Was family history of diabetes or previous gestational diabetes in women been considered? This is particularly important in the analyses considering type 2 diabetes as an outcome.
  • Please use ALT and AST instead of ALAT and ASAT.

Round 2

Reviewer 2 Report

Food job at revising the manuscript. I have no further comments.